# Regulation of alternative splicing at the single-cell level

Lior Faigenbloom[1,‡], Nimrod D Rubinstein[2,†,‡], Yoel Kloog[1], Itay Mayrose[3], Tal Pupko[2,*] & Reuven Stein[1,**]

## Abstract

**Alternative splicing is a key cellular mechanism for generating distinct isoforms, whose relative abundances regulate critical cellular processes. It is therefore essential that inclusion levels of alternative exons be tightly regulated. However, how the precision of inclusion levels among individual cells is governed is poorly understood. Using single-cell gene expression, we show that the precision of inclusion levels of alternative exons is determined by the degree of evolutionary conservation at their flanking intronic regions. Moreover, the inclusion levels of alternative exons, as well as the expression levels of the transcripts harboring them, also contribute to this precision. We further show that alternative exons whose inclusion levels are considerably changed during stem cell differentiation are also subject to this regulation. Our results imply that alternative splicing is coordinately regulated to achieve accuracy in relative isoform abundances and that such accuracy may be important in determining cell fate.**

**Keywords** alternative splicing; evolutionary conservation; inclusion level; single cell; splicing regulation

**Subject Categories** Genome-Scale & Integrative Biology; RNA Biology

**Mol Syst Biol. (2015) 11: 845**

## Introduction

Most mammalian genes are alternatively spliced and produce several protein isoforms, which may have different cellular functions (Matlin *et al*, 2005; Pan *et al*, 2008; Wang *et al*, 2008). This process is considered essential for the evolution of functional complexity in higher eukaryotes (Nilsen & Graveley, 2010). Inclusion levels of alternative exons are determined by the combined action of different splicing factors, which bind to the alternative exons and their flanking intronic regions (FIRs) (Smith & Valcárcel, 2000; Mabon & Misteli, 2005; Wang *et al*, 2008; Nilsen & Graveley, 2010). Specific inclusion levels of various alternative exons are essential for many cellular functions that determine cell fate (Kalsotra & Cooper, 2011). Stochasticity in the regulation of alternative splicing was suggested to lead to heterogeneity in the inclusion levels of alternative exons among individual cells (Waks *et al*, 2011). Dysregulation of inclusion levels has indeed been shown to lead to cellular dysfunctions that result in different human diseases (Venables, 2004, 2006; Tazi *et al*, 2009; Biamonti *et al*, 2012; Chepelev & Chen, 2013). However, what determines the precision of inclusion levels at the individual cell level is still poorly understood.

Studies have shown that FIRs of alternative exons are enriched with splicing regulatory sequences and are evolutionarily conserved to a greater degree than FIRs of constitutive exons (Sorek & Ast, 2003; Barash *et al*, 2010), implying that they may contribute to the regulation of the precision of inclusion levels. To examine this role of FIR conservation, we focused on the cassette type of alternative exons and compiled a dataset of all human cassette exons (Table EV1). To identify cassette exons with highly conserved FIRs, we computed the evolutionary conservation of up to 200 base pairs (bp) upstream and downstream of each cassette exon. This revealed that the evolutionary conservations of the upstream and the downstream FIRs are highly correlated (Spearman's correlation coefficient = 0.69; $P < 10^{-16}$; Fig 1A) and that they both follow a right-skewed distribution, in which only a small fraction of cassette exons have highly conserved FIRs (Fig 1B). We found, moreover, that highly conserved FIRs are enriched with binding motifs of known splicing factors (Tables EV2 and EV3). These observations led us to hypothesize that highly conserved FIRs, which are also associated with splicing regulatory motifs, may play an important role in regulating the precision of their corresponding cassette exon inclusion levels in an individual cell. Therefore, in a homogeneous cell population, we expect that cassette exons with highly conserved FIRs will exhibit significantly lower variability in their inclusion levels than cassette exons with weakly conserved FIRs.

Using single-cell gene expression data from three different cell types, we reveal that splicing of cassette exons is tightly regulated to

1 The Department of Neurobiology, George S. Wise Faculty of Life Sciences, Tel Aviv University, Tel Aviv, Israel
2 The Department of Cell Research and Immunology, George S. Wise Faculty of Life Sciences, Tel Aviv University, Tel Aviv, Israel
3 The Department of Molecular Biology and Ecology of Plants, George S. Wise Faculty of Life Sciences, Tel Aviv University, Tel Aviv, Israel
 *Corresponding author. Tel: +972 6 640 7693; E-mail: talp@tau.ac.il
 **Corresponding author. Tel: +972 3 640 8608; E-mail: reuvens@post.tau.ac.il
 ‡These authors contributed equally to this work
 †Present address: Department of Molecular and Cellular Biology, Harvard University, Cambridge, MA, USA

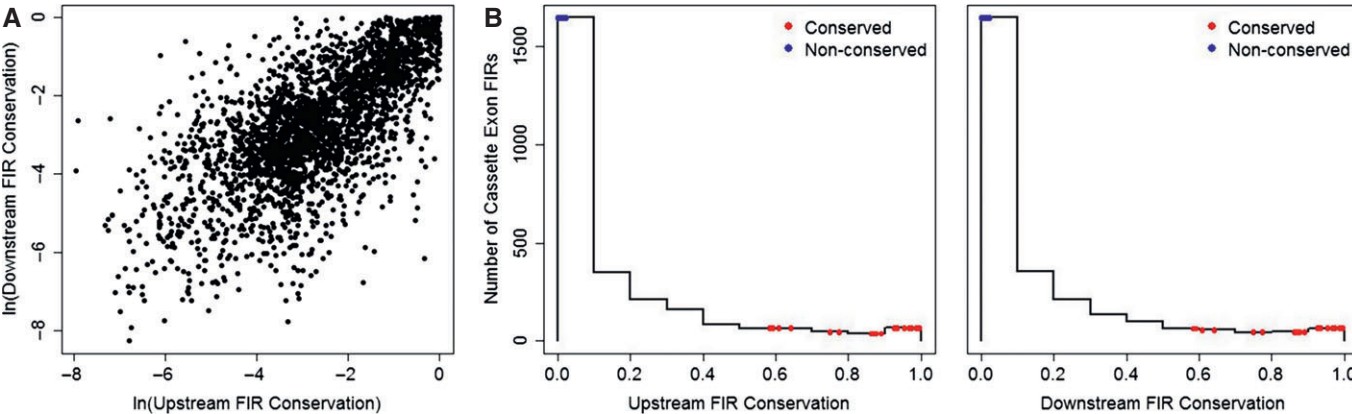

**Figure 1.  Evolutionary conservation at FIRs of cassette exons.**

A   Evolutionary conservations at upstream and downstream FIRs are strongly correlated (Spearman's correlation coefficient = 0.69; $P < 10^{-16}$). Each point represents the average of up to 200-bp position-specific evolutionary conservation scores at upstream and downstream FIRs of the 2,731 cassette exons in our data (Table EV1).

B   Histograms of the evolutionary conservation scores at upstream (left panel) and downstream (right panel) FIRs display a right-skewed distribution. FIR conservations of the 22 conserved and 22 non-conserved cassette exons selected for the single-cell RT–qPCR assay are denoted by red and blue points, respectively.

Source data are available online for this figure.

achieve among-cell precision in relative isoform abundances. We find that this regulation is determined by the conservation level at FIRs of cassette exons, by their inclusion levels, and by the overall expression levels of their corresponding included and skipped isoforms. We further discover that cassette exons whose inclusion levels have been shown to shift dramatically between stem cells and differentiated cells are also subject to such regulation, implying a role of this type of regulation in stem cell differentiation. Our observations thus shed light on what regulates the precision of alternative splicing at the single-cell level.

## Results and Discussion

Splicing of a cassette exon can be viewed as a Bernoulli experiment resulting in either its inclusion or its exclusion (Waks *et al*, 2011; Xiong *et al*, 2011; Wang & Zhou, 2014). Given the properties of a binomial distribution, it is expected that in addition to FIR conservation, dominant exclusion or inclusion levels of cassette

exons (i.e. near 0 or 1, respectively), as well as high expression levels of their corresponding isoforms, will reduce the variability of inclusion levels in individual cells from a homogeneous cell population (illustrated in Appendix Fig S1). To estimate the effect of FIR conservation on the precision of inclusion levels, we measured the expression levels of 44 pairs of included and skipped isoforms obtained from alternative splicing of 22 cassette exons with highly conserved FIRs and 22 cassette exons with weakly conserved FIRs (herein termed conserved and non-conserved groups, respectively) (Fig 1B and Table EV1). Measurements were taken in single cells from homogeneous cell populations. To reduce potential variation, the tested cells were isolated from distinct sub-colonies.

To assess the effects of inclusion and expression levels on the precision of inclusion levels, we used mRNA isolated from total cell populations (herein bulk RNA) to verify that both the conserved and non-conserved groups exhibit broad ranges of inclusion and expression levels (Appendix Fig S2A and B, respectively). In addition, to estimate the possible effects of different cell types on the

**Figure 2.  Single-cell RT–qPCR data.**

A   Heat maps of the expression levels (in $\ln(E_t)$ units) of included and skipped isoforms of the cassette exons. Gene names of the conserved (red) and non-conserved (blue) cassette exons are in rows, and single-cell samples of each of the three cell types are in columns.

B   Heat maps of the estimated inclusion levels of the cassette exons.

C   Dependence of the variance of the cassette exon inclusion levels (*y*-axis) on their mean inclusion levels (*x*-axis) in the RT–qPCR data. Each point represents a cassette exon. This dependence resembles the dependence expected under the assumption that inclusion or exclusion of a cassette exon is a Bernoulli experiment (Appendix Fig S1).

D   FIR conservation increases the precision of inclusion levels. The left panel shows the variance of the transformed inclusion levels as a function of the cassette exon FIR conservation group. The right panel shows the effects (red and blue points) of the two FIR conservation groups on the variance of their inclusion levels as determined by the GLMM analysis along with their standard errors (dashed lines).

E   High expression levels of the included and skipped transcripts increase the precision of their inclusion levels. The left panel shows the variance of the transformed inclusion levels as a function of the cassette exon expression levels. The right panel shows the effect (solid line) of expression level on the variance of inclusion levels as determined by the GLMM analysis, along with their standard errors (dashed lines).

F   Precision of inclusion levels is independent of cell type. The left panel shows the variance of the transformed inclusion levels as a function of cell type. The right panel shows the effects (points) of the three cell types on the variance of inclusion levels as determined by the GLMM analysis, along with their standard errors (dashed lines).

Source data are available online for this figure.

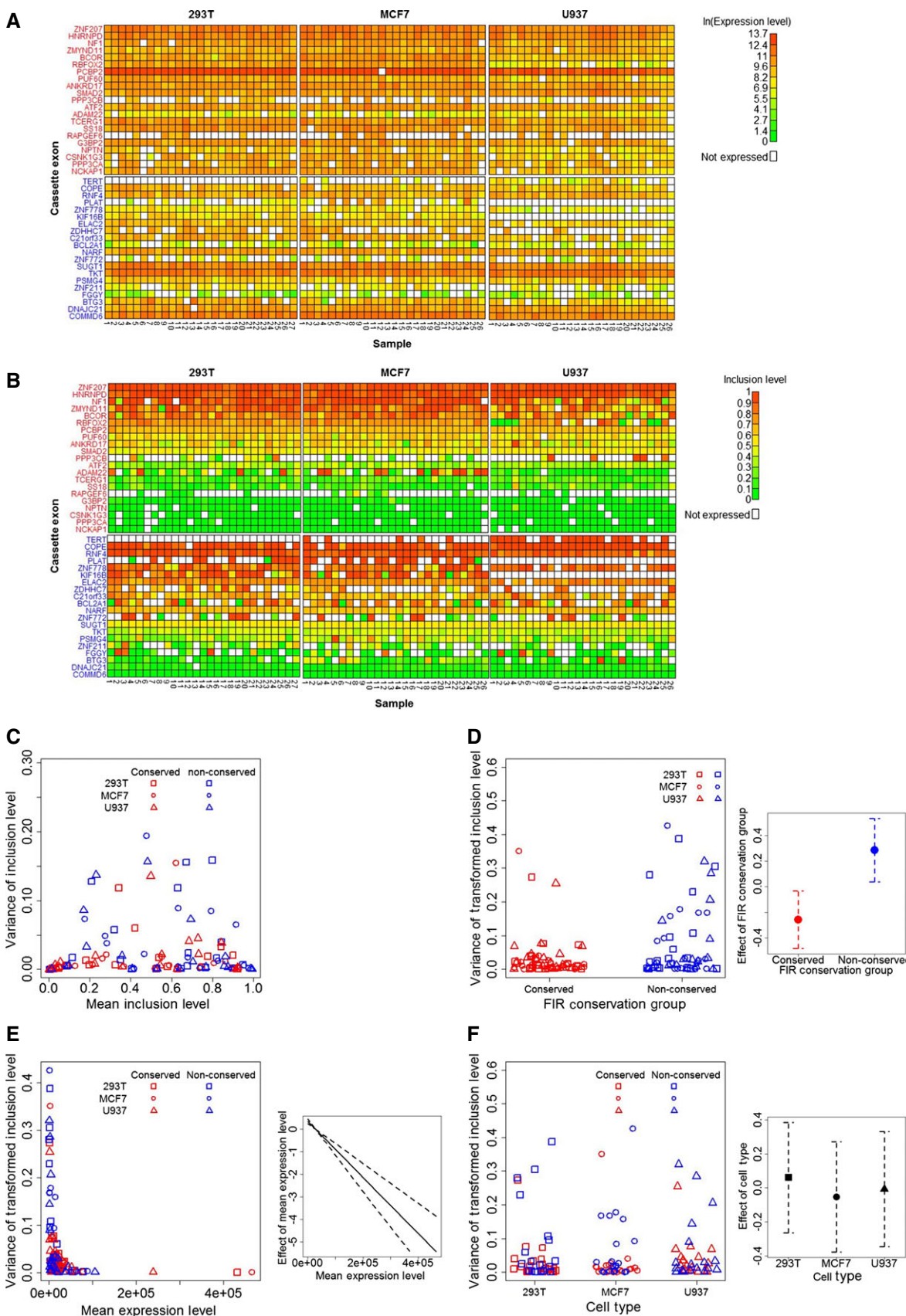

**Figure 2.**

precision of inclusion levels, we selected three different human cell lines (293T, MCF7, and U937), which originate from different tissues and in which all of the selected cassette exons are robustly expressed. Overall, we measured the expression of each of the 88 isoforms from the two FIR conservation groups in 27 single cells for each of the three cell types, using single-cell microfluidic multiplex RT–qPCR (Tables EV4 and EV5).

Subsequent to quality filtering, we estimated the expression levels (as the sum of the expression of included and skipped isoforms) and the inclusion levels of all cassette exons (Fig 2A and B, respectively, and Materials and Methods). Comparing the inclusion levels computed from the total cell populations and those inferred from the single-cell experiment (Materials and Methods) showed no significant differences, indicating that our single-cell experiment did not introduce biases in inclusion levels (Appendix Fig S2C). We additionally ruled out the possibility that the efficiencies of our primer pairs are biased toward either of the FIR conservation groups ($P = 0.46$; see Materials and Methods for details and Appendix Fig S2D). Examination of the relationship between the variance and the mean of inclusion levels across all samples revealed, as expected (Fig 2C), that cassette exons with extreme inclusion levels (near 0 or 1) exhibit lower variability in their inclusion levels than cassette exons with moderate inclusion levels (near 0.5). This demonstrated that the precision of inclusion levels of a cassette exon is strongly affected by the level of its inclusion. We observed that this relationship is not affected by expression levels, as inclusion levels were found not to be correlated with expression levels (Pearson's correlation coefficient = $-0.02$; $P = 0.8$; Appendix Fig S3). Next, we examined the effects of FIR conservation, expression level, and cell type on the variance of inclusion levels. To eliminate the effect of inclusion levels on their variance, we first applied a variance-stabilizing transformation to the estimated inclusion levels (Materials and Methods). Thereafter, to estimate the effect of FIR conservation on the variance of the transformed inclusion levels, while accounting for the effects of expression level (and any difference in expression levels between the conserved and non-conserved groups) and cell type, we fitted a generalized linear mixed effects model (GLMM, Materials and Methods) to these data (see model diagnostics in Appendix Fig S4). This analysis revealed that FIR conservation makes a significant contribution to the model ($P = 0.033$; likelihood ratio test, LRT) and indeed has a significant negative effect on the variance of inclusion levels (estimated coefficient = $-0.54$; standard error (SE) = 0.24, $P = 0.023$; Fig 2D), thus confirming our hypothesis. Furthermore, as expected from the Bernoulli model, the expression level contributes significantly to the model ($P = 1.3 \times 10^{-7}$; LRT) and has a significant negative effect on the variance of inclusion levels (estimated coefficient = $-1.24 \times 10^{-5}$, with SE = $1.8 \times 10^{-6}$; $P = 3.24 \times 10^{-10}$, Fig 2E). Finally, cell type was found not to contribute significantly to the model ($P = 0.91$; LRT, Fig 2F). A control experiment was performed in order to confirm that these results reflect inherent cell-to-cell variability in inclusion levels and not technical factors inherent to our experimental setup. In this control experiment, we used the same microfluidic multiplex RT–qPCR platform to measure the expression levels of the same 88 isoforms in 27 technical replicates, from the same three cell types. To this end, bulk RNA was diluted to a concentration which is lower than the RNA concentration in a single cell, hence allowing a conservative estimate of the technical

variance (see Materials and Methods for details, Tables EV6 and EV7, and Fig EV1 for data). Comparison between the control experiment and the single-cell experiment revealed that the expression levels are significantly lower in the control experiment than in the single-cell experiment ($P < 2 \times 10^{-16}$), which is expected since in the control experiment we diluted RNA to below single-cell concentrations. While the inclusion levels of all cassette exons in the control experiment were found to vary significantly more than expected under the binomial distribution ($P < 0.05$ for all cassette exons; chi-square test), indicating a baseline technical noise, the variance of inclusion levels in this experiment was, nevertheless, significantly lower than the variance of inclusion levels in the single-cell experiment ($P = 0.03$). Similar to the single-cell experiment, a relationship between the variance of inclusion levels and their mean, which is expected under the Bernoulli model of alternative splicing (Fig EV1C), was observed in the control experiment. Modeling the variance of inclusion levels (see model diagnostics in Appendix Fig S5) detected a significant contribution of mean expression levels ($P = 0.007$; LRT) with a significant negative effect (estimated coefficient = $-4.02 \times 10^{-6}$, with SE = $5.49 \times 10^{-7}$; $P = 6.75 \times 10^{-7}$, Fig EV1D), which again is expected under the Bernoulli model of alternative splicing. However, in contrast to inclusion and expression levels, FIR conservation was not found to have a significant effect on the variance of inclusion levels ($P = 0.076$; LRT). These results indicate that the precision of inclusion levels of a cassette exon is significantly determined by three factors: its FIR conservation, its inclusion level, and the expression levels of both its included and its skipped isoforms, each of which is a significant contributor to the precise balance between the alternative isoforms, even when the contribution of the other two factors is taken into account. Furthermore, the lack of a significant effect of cell type on the variance of inclusion levels suggests that regulation of the precision of inclusion levels is a general phenomenon that is not cell type specific.

Having established a relationship between FIR conservation and the precision of inclusion levels, we performed a Gene Ontology (GO) analysis (Eden et al, 2009) to determine the biological processes in which genes harboring cassette exons with conserved FIRs are enriched. This analysis revealed a significant enrichment of developmental processes (Table EV8), suggesting that the precision of cassette exon inclusion levels may play a key role in cells undergoing processes such as differentiation in stem cells. Shifts in inclusion levels of various alternative exons between stem cells and differentiated cells (referred to here as differentiation-switched cassette exons) have indeed been proposed as one of the mechanisms controlling stem cell differentiation (Gabut et al, 2011; Han et al, 2013; Venables et al, 2013; Irimia et al, 2014). We therefore tried to determine what affects the precision of cassette exon inclusion levels in stem cells, focusing on differentiation-switched cassette exons shown to undergo substantial changes in their inclusion levels during stem cell differentiation (Han et al, 2013). We investigated this by estimating inclusion and expression levels (Materials and Methods) from a single-cell RNA-seq data generated from two populations of undifferentiated human embryonic stem cells (hESCs) (Yan et al, 2013). One population, denoted by P0, consists of eight samples of cells that were not permitted to undergo any passages, and the second population, denoted by P10, consists of 26 samples from cells that underwent ten passages (Fig 3A and B

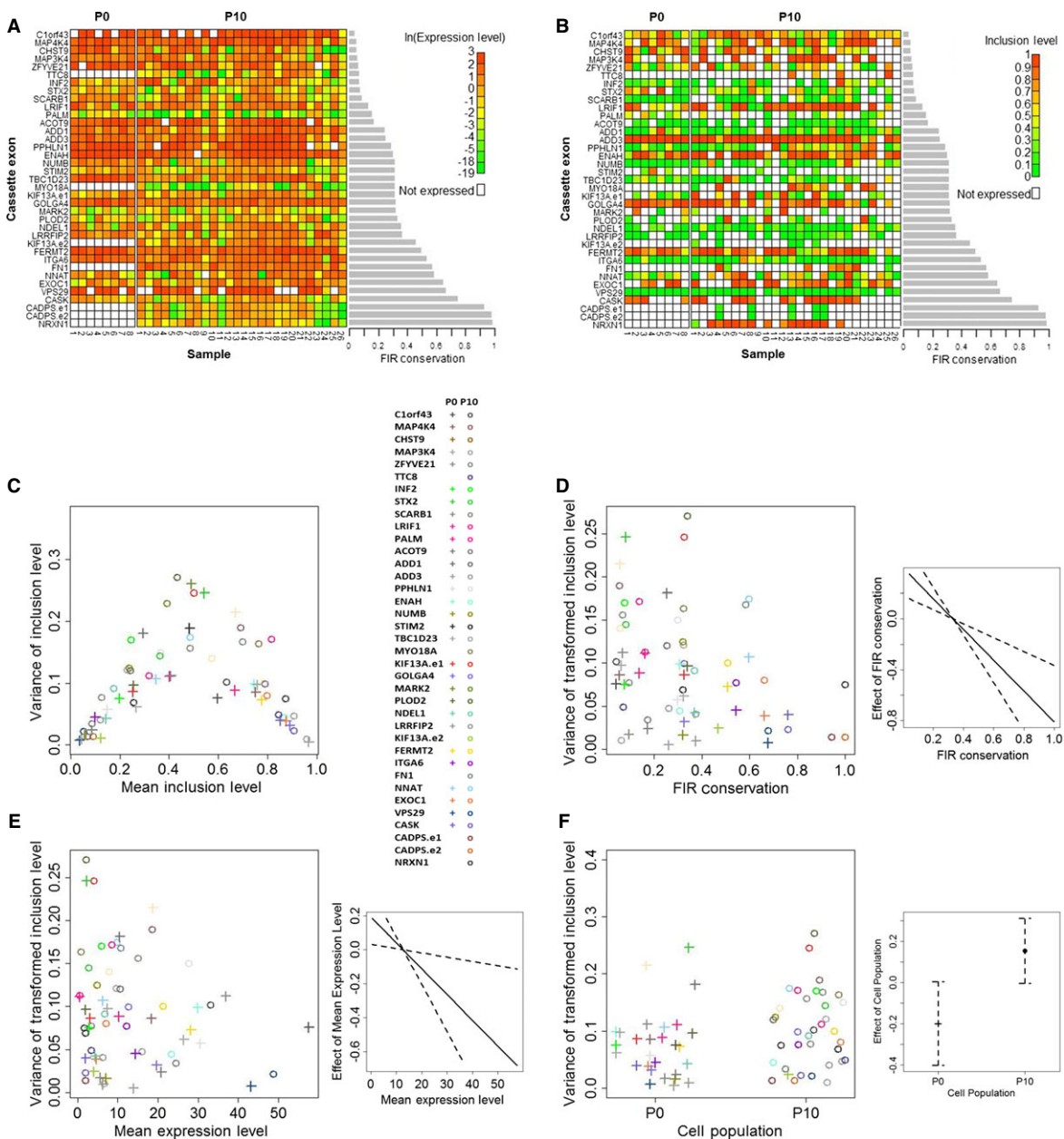

**Figure 3. Single-cell human embryonic stem cell data.**

A   Heat maps of the expression levels (in ln(*FPKM*) units) of the genes harboring the differentiation-switched cassette exons. Gene names of the differentiation-switched cassette exons are in rows (the CADPS and KIF13A genes harbor two differentiation-switched cassette exons which are labeled CADPS.e1 and CADPS.e2 and KIF13A.e1 and KIF13A.e2), and single-cell samples of each of the two cell populations are in columns.

B   Heat maps of the estimated inclusion levels of the differentiation-switched cassette exons.

C   Dependence of the variance of differentiation-switched cassette exon inclusion levels (*y*-axis) on their mean inclusion levels (*x*-axis). The color code for each cassette exon is given in the right panel. This dependence resembles the expected dependence under the assumption that inclusion or exclusion of a cassette exon is a Bernoulli experiment (Appendix Fig S1).

D   FIR conservation increases the precision of inclusion levels. The left panel shows the variance of the transformed inclusion levels as a function of the differentiation-switched cassette exon FIR conservation. The right panel shows the effect (solid line) of FIR conservation of differentiation-switched cassette exons on the variance of their inclusion levels as determined by the GLMM analysis along with their standard errors (dashed lines).

E   High expression levels of transcripts harboring the differentiation-switched cassette exons increase the precision of their inclusion levels. The left panel shows the variance of the transformed inclusion levels as a function of the differentiation-switched cassette exon expression levels. The right panel shows the effect (solid line) of expression level on the variance of inclusion levels as determined by the GLMM analysis, along with their standard errors (dashed lines).

F   Cell population has a negligible effect on the precision of inclusion levels. The left panel shows the variance of the transformed inclusion levels as a function of the cell population. The right panel shows the effects (points) of the two cell populations on the variance of inclusion levels as determined by the GLMM analysis, along with their standard errors (dashed lines).

Source data are available online for this figure.

and Table EV9). As both expected and observed in the RT–qPCR data (Fig 2C), the variance of inclusion levels of the differentiation-switched cassette exons showed strong dependence on their mean inclusion levels, with lower variances near mean inclusion levels close to 0 and 1 and higher variances near mean inclusion levels close to 0.5 (Fig 3C). After eliminating dependence on the inclusion levels using the variance-stabilizing transformation (Materials and Methods), as with the RT–qPCR data, we observed that variability of inclusion levels declines with increasing FIR conservation of differentiation-switched cassette exons (Fig 3D). To quantify this relationship, we fitted a GLMM (Materials and Methods) to these data (see model diagnostics in Appendix Fig S6). This analysis revealed that FIR conservation contributes significantly to the model ($P = 0.002$; LRT) and has a corresponding significant negative effect on the variance of inclusion levels (estimated coefficient = $-2.03$, SE = 0.26, $P = 2.87 \times 10^{-4}$; Fig 3D). Expression level was also found to contribute significantly to the model ($P = 0.042$; LRT), with a significant negative effect on the variance of inclusion levels (estimated coefficient = $-0.015$, SE = 0.007; $P = 0.0027$, Fig 3E). Although both cell populations originated from the same embryonic state, the P10 population could have begun to differentiate and thus might show a different pattern of precision in inclusion levels. Therefore, we chose to distinguish between them in the analysis. This, however, was not found to be the case as cell population was not found to contribute significantly to the model ($P = 0.066$; LRT), indicating that it does not significantly affect the precision of inclusion levels in these data (estimated coefficient = 0.35, SE = 0.18; $P = 0.059$, Fig 3F). These results therefore corroborate our findings from the single-cell RT–qPCR analysis, again indicating that the precision of inclusion levels of cassette exons is significantly affected by their FIR conservation, as well as by their inclusion levels, and the expression levels of their corresponding isoforms.

The finding that differentiation-switched cassette exons are subject to this type of regulation in stem cells suggests an important role for precision of their inclusion levels in stem cell differentiation. This may allow stem cells to tightly maintain their stem cell state, preventing leakage toward differentiation in the absence of a differentiation signal. It may also provide a mechanism by which differentiating stem cells can faithfully adhere to a developmental axis, thus avoiding drift toward abnormal differentiation pathways (Chepelev & Chen, 2013). As an example, the cassette exon with the most highly conserved FIRs in our data is exon AS4 of the NRXN1 gene (Table EV1). The NRXN gene family mediates synaptogenesis (Li et al, 2007) and has been implicated in complex neurodevelopmental and neuropsychiatric disorders, such as autism spectrum disorders and schizophrenia (Südhof, 2008; Reichelt et al, 2012). Alternative splicing of AS4 modulates the interaction of NRXN1 with several ligands (Li et al, 2007; Südhof, 2008) with different synaptic functions (Boucard et al, 2005; Chih et al, 2006; Iijima et al, 2011). Remarkably, AS4 is a differentiation-switched cassette exon that shows high precision in its inclusion levels in the hESC data (Fig 3D and Table EV9).

Taken together, our results suggest that splicing of certain cassette exons is subject to tight regulation to facilitate the precision of their inclusion levels among cells in a homogeneous population. Our main finding indicates that this precision in inclusion levels is under strong selective pressure, as manifested by the conservation of FIRs of such cassette exons. Our results also emphasize the

Bernoulli nature of alternative splicing by revealing that high precision of inclusion levels can be achieved through high expression levels and/or extreme inclusion levels. In line with our findings suggesting that the relative abundances of the included and the skipped isoforms are tightly regulated, aberrant variations in isoform abundances have been shown to lead to abnormal cellular functions, changes in cell fate, and cancer (Venables, 2004; David & Manley, 2010; Biamonti et al, 2012). Ensuring high precision in inclusion levels of cassette exons may therefore be a possible mechanism for avoiding such cellular malfunctions.

# Materials and Methods

### Human cassette exon data

The tables of the human genome hg19 assembly RefSeq (Pruitt et al, 2007) transcriptome (knownToRefSeq) and the University of California Santa Cruz (UCSC) genome browser (Kuhn et al, 2013) alternative events (knownAlt) were downloaded from the UCSC genome browser. The cassette-type alternative exons from the knownAlt table were selected and further filtered to retain only cassette exons that are annotated in the knownToRefSeq table. Only cassette exons, for which we found at least one pair of included and skipped transcripts in the knownToRefSeq table, were further selected. To compute FIR conservation at 200 bp upstream and downstream positions of all cassette exons, we used the UCSC genome browser phastCons46way table, which provides a normalized position-specific evolutionary conservation score (in the range of [0,1]), based on a multiple sequence alignment of 33 placental mammalian genomes. To avoid using FIR sequences that are part of a coding region in other transcripts (according to the more extensive transcriptome annotation—the UCSC genome browser knownGene table), these overlapping sequences were not included in the analyzed FIR sequences. Cassette exons harboring more than 100-bp overlap with a coding region in either their upstream or their downstream FIR were not included in the analysis. As a result of all these filtering steps, 2,731 cassette exons were retained (Table EV1).

### Enrichment analysis of splicing factor binding motifs at cassette exon flanking intronic regions

To determine whether cassette exons with conserved FIRs are enriched in splicing factor binding motifs, we first searched for FIR-enriched k-mers using DRIMust (Leibovich et al, 2013), a tool for identifying short motifs in a ranked list of nucleic acid sequences. We performed this analysis independently for upstream and downstream FIR sequences ranked by their conservation scores, using a motif length range of 5–11 bp, which covers the range of annotated binding motifs of known splicing factors (Cook et al, 2011; Ray et al, 2013) (Tables EV2 and EV3). To avoid detecting enrichment of sequences that are part of the branch site, polypyrimidine tract, or 3′ and 5′ splice sites, these splicing sequences were predicted with SROOGLE (Schwartz et al, 2009) and removed from the inspected FIR sequences. Next, we computed the similarity of all the significantly enriched k-mers reported by DRIMust (false discovery rate (FDR)-corrected (Benjamini & Hochberg, 1995) $P < 0.05$ in both

cassette exon groups) to a list of motifs of known splicing factors (Cook *et al*, 2011; Ray *et al*, 2013) (Tables EV2 and EV3), using the TOMTOM tool (Gupta *et al*, 2007) and retaining all hits with FDR-corrected $P < 0.05$.

### Gene ontology enrichment analysis of cassette exons with conserved flanking intronic regions

To determine whether cassette exons with conserved FIRs are enriched in specific biological processes, we performed a GO enrichment analysis using the GOrilla tool (Eden *et al*, 2009). The input to GOrilla was the list of genes harboring the cassette exons in our data (Table EV1) ranked by their mean upstream and downstream FIR evolutionary conservation scores. The output of GOrilla is GO terms that are enriched in the genes at the top of the list. Genes harboring more than a single cassette exon were removed from this analysis.

### Single-cell RT–qPCR assay

#### Selection of cassette exons

The following criteria were used for selecting the cassette exons for the single-cell RT–qPCR assay. First, to estimate the effect of FIR conservation on the precision of inclusion levels, we selected two groups each containing 22 cassette exons, where one group included cassette exons with highly conserved FIRs and the other included weakly conserved FIRs (Fig 1B). To account for the possible effects of expression level and inclusion levels on the precision of inclusion levels, we also required that the expression levels of included and skipped isoforms of the selected cassette exons and their inclusion levels should span a wide range (Appendix Fig S2A and B). Finally, we required that each of our selected cassette exons be expressed in the three different cell types, according to the BioGPS database (Su *et al*, 2004; Wu *et al*, 2009). This was further verified by RT–qPCR experiments (Table EV10). The list of the selected cassette exons is provided in Table EV1.

#### Cell lines and sub-cloning

Human embryonic kidney 293T and breast cancer MCF7 cells were grown in DMEM with 10% fetal calf serum (FCS). Human U937 cells (established from a histiocytic lymphoma) were grown in RPMI 1640 with 10% FCS. Homogeneous cell populations were then obtained by two cycles of sub-cloning. For each cell line, single cells were seeded at a low density and allowed to propagate until single colonies were generated. Several of these colonies were then selected and dissociated to single cells. Small portions of these cells were seeded again at low density to generate a second round of sub-clones. The remaining cells were reseeded, grown, and harvested to prepare bulk RNA samples of the total cell population. Sub-clones in the second round were allowed to propagate for ~10 divisions to yield clones of roughly 1,000 cells. Cells from single sub-clones were then dissociated and sorted into a 384-well plate by flow cytometry to obtain a single cell per well, and 27 single cells were subjected to the single-cell RT–qPCR assay. The bulk RNA samples were used to calibrate the primers of isoforms that were used in the single-cell microfluidic multiplex RT–qPCR analysis.

#### Single-cell isolation by flow cytometry

Cells from the second round of sub-cloning were trypsinized, washed in phosphate-buffered saline (PBS), and resuspended in PBS. Cells were sorted by a SORP FACSAria II cell sorter, which was calibrated with a sample of "pooled" cells to ensure that the deposited cells had the same forward- and side-scatter settings. From each cell line, 59 cells were sorted directly into 384-well plates, where each well contained 5 μl of CellsDirect Reaction Mix (Invitrogen) and 0.05 units of SUPERase In RNase inhibitor. The plates were then centrifuged and stored at −80°C until they were used to generate cDNA libraries.

#### Generation of cDNA libraries from single cells

cDNAs from each of the sorted single cells were prepared according to the manufacturer's protocol (Fluidigm). Briefly, first-strand cDNA was synthesized using STA primer pool mix (Table EV11), followed by 18 cycles of sequence-specific pre-amplification and exonuclease I treatment. Samples were then diluted five-fold and mixed with SsoFast EvaGreen Supermix with low ROX (Bio-Rad Laboratories) and DNA Binding Dye Sample Loading Reagent (Fluidigm) before being loaded into the 96.96 Dynamic Array IFC (Fluidigm).

#### Verification of the amount and quality of single-cell cDNA

To verify that all of the abovementioned pre-amplified cDNAs were originated from single cells and that their amounts were similar, we examined the expression levels of three housekeeping genes (HKGs): GAPDH, RPS13, and RPL29 by RT–qPCR performed on the pre-amplified cDNAs from each of the 59 single-cell samples. Only single-cell samples that expressed similar levels of the HKGs were chosen for the microfluidic multiplex RT–qPCR assay. To this end, for each HKG, we computed the median expression level across all single-cell samples and then the deviation of expression of each HKG in each single-cell sample from its respective median expression. Finally, we ranked all single-cell samples according to the maximal distance of HKGs from their median expression in ascending order and selected the top 27, from each of the three cell types for the RT–qPCR assay.

#### Primer design and calibration

Using the NCBI primer blast, we designed primers that specifically differentiate between the included and skipped isoforms, with respect to each of the selected cassette exons. For each cassette exon, we designed three primers: one reverse primer, which was used to amplify both isoforms and to convert RNA to cDNA, and two forward primers, each used to specifically amplify one of the two isoforms (Table EV11). The reverse primers were designed to localize the constitutive exon downstream of the cassette exons. The forward primer amplifying the included isoform was designed to localize to the junction of the cassette exon and its downstream constitutive exon. If a primer could not be designed according to these requirements, a corresponding primer was designed so that it localized only to the cassette exon, thus still allowing specific recognition of the included isoform. The forward primer amplifying the skipped isoform was designed to localize to the junction of the upstream and the downstream constitutive exons. To prepare ×10 pooled primer mix, each primer pair (50 μM) was diluted to a concentration to 500 nM, by combining 2 μl from each primer pair with 18 μl of DNA suspension buffer.

Primer calibration was performed by RT–qPCR. For each cDNA sample, we used five different five-fold serial dilutions (25, 5, 1, 0.2, and 0.04 ng cDNA per reaction). Each reaction was checked for quality (by visual inspection of the multicomponent curve), primer specification (by visual inspection of the peak of the dissociation curve), standard curve slope of cDNA dilutions, and the overall efficiency of amplification (computed using the Viia7 software). Only primers that exhibited specificity for their corresponding isoforms with an efficiency of $100 \pm 20\%$ were selected for the single-cell RT–qPCR assay. Finally, we tested that our primer efficiencies are not significantly biased toward one of the FIR conservation groups. To this end, we used a linear mixed effects model where the response was defined as the primer efficiency, FIR conservation group, cell type, and isoform (included or skipped) were defined as fixed effects, and the cassette exon was defined as a random effect. According to this model, FIR conservation group did not have a significant effect on the primer efficiency ($P = 0.46$; LRT, Appendix Fig S2D), thus ruling out a technical bias due to different primer efficiencies between the two FIR groups.

*Single-cell RT–qPCR assay*
The 96.96 Dynamic Array IFC was loaded with cDNAs from each of the 27 selected single cells for each of the three cell types, as well as with three no-template controls (NTCs). We loaded 88 primer pairs corresponding to the 44 pairs of included and skipped isoforms as well as primer pairs for the three HKGs (used as positive controls) and no-primer controls (NPCs), both loaded in duplicate. The 96.96 Dynamic Arrays IFC was then loaded on a BioMark System and run for 30 PCR cycles ($C_{max} = 30$), according to the manufacturer's recommendations.

*Control RT–qPCR assay*
cDNA samples were synthesized from the bulk RNA samples, from each of the three cell lines. Each bulk RNA sample was then diluted to ~1 ng per reaction, followed by 6 more 3.3-fold serial dilutions (the lower concentration was ~0.001 ng per reaction). These diluted RNA samples were then used to prepare cDNA samples following the same protocol we used to prepare the single-cell cDNA samples (using SuperScript® III One-Step RT-PCR System with Platinum® *Taq* DNA Polymerase by Invitrogen). To choose pre-amplified cDNAs that have expression levels similar to the cDNAs that were used in the single-cell experiment, we examined the expression levels of three housekeeping genes (HKGs): GAPDH, RPS13, and RPL29 as well as one alternative isoform (ANKRD17 skipped isoform) by RT–qPCR performed on the pre-amplified cDNAs from each dilution. For the control microfluidic multiplex RT–qPCR experiment, we chose the cDNA dilution that the resulted expression levels of these four control genes were the closest but lower than the mean expression level exhibited by the single cells as obtained by RT–qPCR analysis. We subsequently divided these diluted cDNA samples from each cell line to 27 equal samples (replicates) and loaded them into the 96.96 Dynamic Array IFC. In addition, the 96.96 Dynamic Array IFC was loaded with three no-template controls (NTCs): 88 primer pairs corresponding to the 44 pairs of included and skipped isoforms, primer pairs for the three HKGs loaded in duplicate, and no-primer control (NPC), also loaded in duplicate. The 96.96 Dynamic Arrays

IFC was then loaded on a BioMark System and run for 30 PCR cycles ($C_{max} = 30$), according to the manufacturer's recommendations.

**Single-cell RT–qPCR data analysis**

*Filtering of samples with unsuccessful reactions*
Any $C_t$ call that was marked as "failed" by the Fluidigm Real-Time PCR Analysis Software was eliminated. For this, we used the following criteria: $C_t$ quality > 0.65; peak ratio ($T_m$ peak detected within the $T_m$ detection range/total detection) > 0.8.

*Filtering of samples with cDNA amplification failure*
To account for the possibility of cDNA amplification failure, we followed the procedure described in Livak *et al* (2013) and defined

$$I_{x,r} = \begin{cases} 1 \text{ if } Ct < C\max \\ 0 \quad \text{else} \end{cases},$$

which indicates whether expression was observed in a well containing a primer set $x$ and a single-cell cDNA sample $r$. We defined the probability of expression for each targeted isoform as $I_x = \sum_{r \in R} I_{x,r}/R$, where $R$ denotes the number of single-cell cDNA samples. Next, we computed a failure-of-expression penalty for each well as

$$s_{x,r} = \begin{cases} I_x \text{ if } Ct = C\max \\ 0 \quad \text{else} \end{cases}.$$

Finally, we computed the failure-of-expression score for each single-cell cDNA sample $r$ as $S_r = \sum_{x \in X} s_{x,r}$. We subsequently removed two single-cell cDNA samples for which outlying $S_r$ values were clearly observed (Appendix Fig S7).

*Filtering samples with expression below the limit of detection*
To eliminate samples that represent noise, we computed the limit of detection (LOD). According to the manufacturer's guidelines, $E_t$ values equal to or lower than 8 (where $E_t = C_{max} - C_t$) may not reliably represent true expression. For a given isoform whose typical $E_t$ value is higher than 8, noisy samples would be expected to appear as outliers of the distribution of the reliable samples. To detect such outliers for each isoform, we determined the LOD by iteratively increasing it, starting from the lowest observed $E_t$ up to $E_t = 8$, and filtering any sample identified as a statistically significant outlier by Grubbs' test for outliers (Grubbs, 1969), which is designed to detect outliers in a bell-shaped distribution, the typical shape of the $E_t$ distribution across the single-cell RT–qPCR samples (Appendix Fig S8).

**Calculation of cassette exon inclusion levels and expression levels of included and skipped isoforms for the RT–qPCR data**

We used $2^{Et} - 1$ as a proxy for expression level. Since all our primer pairs were calibrated for more than 90% efficiency, we assume that $2^{Et} - 1$ reliably approximates true expression levels. We estimated the inclusion level of a cassette exon ($p$) as $\hat{p} = \frac{2_I^{Et} - 1}{2_I^{Et} + 2_S^{Et} - 2}$, where $2_I^{Et} - 1$ and $2_S^{Et} - 1$ denote the expression levels of the included and the skipped isoforms, respectively. Accordingly, $2_I^{Et} + 2_S^{Et} - 2$ was the expression level used in the analysis of these data. $\hat{p}$ is therefore the maximum-likelihood estimate of the inclusion probability (or inclusion level) $p$, which is the success

probability parameter of a binomially distributed random variable for which $2_I^{Et} - 1$ successes were observed out of $2_I^{Et} + 2_S^{Et} - 2$ trials.

### Filtering cassette exons with no evidence of alternative splicing

Any cassette exon that was either only included or only skipped in all its samples which passed the previous filtering steps, in a given cell type, was additionally filtered since this reflects the lack of evidence of alternative splicing in the respective cell type.

### Variance-stabilizing transformation of inclusion levels

To eliminate the dependence of variance of estimated inclusion levels on the estimated inclusion levels ($\hat{p}$), we applied the arcsin $\left(\sqrt{\hat{p}}\right)$ variance-stabilizing transformation (Sokal & Rohlf, 1995) to all values of $\hat{p}$. We used the sample mean and sample variance as estimates of the across-sample mean and variance of both $\hat{p}$ and arcsin $\left(\sqrt{\hat{p}}\right)$.

### Human embryonic stem cell single-cell RNA-seq data

We downloaded all eight samples of the 0 passages (P0) and 26 samples of the 10 passages (P10) from the raw RNA-seq data of single-cell hESCs populations, generated by Yan *et al* (2013) (Gene Expression Omnibus accession: GSE36552). Data were subjected to quality filtering using the FastQC software (http://www.bioinformatics.babraham.ac.uk/projects/fastqc).

### Calculation of cassette exon inclusion and expression levels of included and skipped isoforms for the hESC RNA-seq data

We aligned all hESC single-cell read data to the hg19 human genome assembly along with the RefSeq splice junction annotation (Pruitt *et al*, 2007) using the STAR RNA aligner (Dobin *et al*, 2013) with default parameters. We quantified gene expression levels (reported in fragment per kilobase per million sequenced reads units, FPKM) using Cufflinks (Trapnell *et al*, 2010), with default parameters (Table EV9), and used them as proxy for the sum of expression levels of the included and skipped isoforms.

To estimate the inclusion levels of all cassette exons expressed in the hESC data (Table EV9), we used MISO (Katz *et al*, 2010) (with default parameters), which applies a Bayesian model for this task. As a result, for each cassette exon in each single-cell RNA-seq sample, MISO computes samples from the posterior distribution of the inclusion level (i.e. $\hat{p}^s$, which denotes sample $s$ of $S$ samples from the posterior distribution of the inclusion level $p$). The variance of this posterior distribution represents uncertainty in the estimated inclusion level (i.e. measurement error), which stems from the mapping ambiguity of the short-read RNA-seq data.

### Estimating the variance of inclusion levels for the hESC RNA-seq data

Similar to the analysis of the single-cell RT–qPCR data, we applied the arcsin $\left(\sqrt{\hat{p}}\right)$ transformation to every sample $\hat{p}^s$ of $S$ samples from the posterior distribution of $p$. To obtain estimates of the variance of arcsin $\left(\sqrt{\hat{p}}\right)$ of a specific cassette exon across $R$ single-cell RNA-seq samples, from each $r \in R$ RNA-seq sample we randomly drew a sample from the posterior distribution of arcsin $\left(\sqrt{\hat{p}}\right)$ and subsequently computed the sample variance over these posterior samples. That is,

$$s^2_{\arcsin\left(\sqrt{\hat{p}}\right)}{}^s = \frac{1}{R-1}\sum_{r \in R}\left[\arcsin\left(\sqrt{\widehat{p}_r^{s_r}}\right) - \frac{1}{R-1}\sum_{r \in R}\arcsin\left(\sqrt{\widehat{p}_r^{s_r}}\right)\right],$$

where $s_r$ denotes a specific sample draw $s$ for single-cell RNA-seq sample $r$, and therefore, $s^{s2}_{\arcsin\left(\sqrt{\hat{p}}\right)}$ approximates a sample $s$ from the posterior distribution of the variance of arcsin $\left(\sqrt{\hat{p}}\right)$ across $R$ single-cell RNA-seq samples. We repeated this process many times (where the draws are performed with replacement) in order to obtain samples from the posterior distribution of the variance of arcsin $\left(\sqrt{\hat{p}}\right)$ across $R$ single-cell RNA-seq samples.

### Fitting a generalized linear mixed effects model to the variance of cassette exon inclusion levels

For both the single-cell RT–qPCR data and the hESC RNA-seq data, we fitted a generalized linear mixed effects model (GLMM) for estimating the effects of FIR conservation, expression levels, and cell type or cell population, on the variance of arcsin $\left(\sqrt{p}\right)$. Specifically, we used the generalized additive models for location, scale, and shape (GAMLSS) (Rigby *et al*, 2005) implemented in the gamlss R package.

For the single-cell RT–qPCR data, we defined the response as the sample variance of arcsin $\left(\sqrt{p}\right)$ of a specific cassette exon across all of its single-cell cDNA samples from a specific cell type (i.e. $s^2_{\arcsin\left(\sqrt{p}\right)}$). In both RT–qPCR and hESC data, we discarded any cassette exon for which less than three single-cell samples remained subsequent to all filtering steps. The number of samples of each cassette exon in the single-cell RT–qPCR and hESC analyses is provided in Tables EV5 and EV9, respectively, and is presented as histograms in Appendix Fig S9A and B, respectively. The mean expression level across the corresponding single-cell cDNA samples was defined as a continuous fixed effect, and the conservation group (i.e. conserved or non-conserved) and the cell type (i.e. 293T, MCF7, or U937) were defined as categorical fixed effects. Lastly, the cassette exon was defined as a random effect. We fitted GAMLSS using the gamma family distribution which applies a log link function to both the mean response and the variance of the errors. We accounted for the different number of samples in the different FIR conservation groups and cell types (which affects the sample variance, i.e. the response) by weighing the response by the corresponding number of samples divided by the mean number of samples across all responses. Model fit was assessed with diagnostic plots (Appendix Figs S4 and S5), where outliers, defined as observations which residuals from the regression fit were found to be significant outliers according to the Grubbs' test (Grubbs, 1969), were removed. For each specified effect, to assess whether its inclusion in the model is significantly justified, we performed a likelihood ratio test (LRT) between a model that includes it and a nested model that does not. In addition, the estimated coefficient of each effect in the GAMLSS fit is computed a *P*-value, obtained from a *t*-distribution with respective degrees of freedom in the model, where the statistic is the estimated coefficient of the effect divided by its standard error.

In order to assess whether the difference in the numbers of samples between the two FIR conservation groups creates an artificially significant effect on the variance of inclusion levels, we performed the following simulation study. We used the experimental design of the RT–qPCR data, that is, two groups of cassette exons from three different treatments (cell types) with sample

sizes identical to those in the post-filtering RT–qPCR data. For each cassette exon, we drew an inclusion level (denoted as $\pi$) uniformly between 0 and 1 along with a dispersion factor (denoted as $\delta$) drawn uniformly between 0.01 and 0.05. Then, for each cassette exon in each sample, we drew a dispersed inclusion level uniformly from the [max $(\pi-\delta, 0)$, min $(\pi+\delta, 1)$] interval, thus reflecting random dispersion in inclusion levels across samples. We then drew an expression level of the included isoform for each cassette exon from a binomial distribution given its simulated dispersed inclusion level and expression level from the real data (i.e. the number of successes given the success probability and number of trials). Following that, we estimated the inclusion level in each sample (i.e. $\hat{\pi}$) as the number of included isoforms divided by the expression level, and transformed them (i.e. arcsin $(\sqrt{\hat{\pi}})$). Finally, we fitted the same GLMM to these simulated data and computed the LRT $P$-value of the group effect. Out of 1,000 such simulations, the fraction of simulations with a group-effect LRT $P < 0.05$ was found to be 6%, thereby confirming that the significant effect of FIR conservation group, observed in the real data, is not an artifact of the different numbers of samples between the two FIR conservation groups.

For the hESC RNA-seq data, the same model was fitted to the response defined as a sample from the posterior distribution of $s^2_{\text{arcsin}(\sqrt{\hat{p}})}$ (i.e. $s^2_{\text{arcsin}(\sqrt{\hat{p}})}{}^s$). The mean expression-level fixed effect was defined as the mean FPKM of the gene harboring the cassette exon across the single-cell RNA-seq samples (computed as explained above), and FIR conservation was defined as a continuous fixed effect. Cell population (i.e. P0 or P10) was defined as a categorical fixed effect similar to the cell-type fixed effect in the single-cell RT–qPCR data, and the cassette exon was similarly defined as a random effect. Again, each response was weighed by its corresponding number of samples divided by the mean number of samples across all responses. We fitted this model for each posterior sample $s^2_{\text{arcsin}(\sqrt{\hat{p}})}{}^s$, thereby obtaining posterior samples of fitted models. In contrast to the RT–qPCR data, in these data the cassette exons with low FIR conservation have more samples than cassette exons with high FIR conservation (Table EV9 and Appendix Fig S9). Therefore, the significant effect of FIR conservation on the variance of inclusion levels is unlikely to arise due to these differences in the numbers of samples. We note that limiting this analysis to a minimum of ten single-cell samples per cassette exon (which means using only the P10 population) essentially did not affect our results.

In all figures and analyses for the hESC RNA-seq data, we used the mean of posterior sample variance ($\sum_{s \in S} s^2_{\text{arcsin}(\sqrt{\hat{p}})}{}^s/S$). The 95% credible intervals for these posterior samples for each estimated effect coefficient, their standard error, and the corresponding $P$-values as well as the $P$-values for the model selection tests are provided in Table EV12.

### Comparison of the control and single-cell RT–qPCR experiments

To compare the expression levels between the control and the single-cell RT–qPCR experiments, we used a GLMM where we defined the response as the natural log fold change between the means of expression levels across the samples of the two experiments. FIR conservation group and cell type were defined as fixed effects, and cassette exon was defined as a random effect. To compare the variance of inclusion levels between the control and the single-cell RT–qPCR experiments, we used a GLMM where we defined the response as the natural log fold change between the variances of arcsin-transformed inclusion levels across samples of the two experiments. FIR conservation group and cell type were defined as fixed effects as well as the natural log fold change of the mean expression levels between the two experiments. Cassette exon was defined as a random effect.

In order to test whether the variance of inclusion levels in the control experiment is within the range expected under the binomial distribution, for each cassette exon we performed a chi-square test assuming that the null inclusion level is the mean inclusion level over all samples.

In order to estimate the effects of FIR conservation group, mean expression levels, and cell type on the variance of inclusion levels in the data of the control experiment, we fitted the same GLMM used for the analysis of single-cell RT–qPCR data.

### Data availability

The RT–qPCR primary data are available in Tables EV4 and EV5. The hESC referenced data were generated by Yan *et al* (2013) and were downloaded from the Gene Expression Omnibus under accession GSE36552.

**Expanded View** for this article is available online.

### Acknowledgements

We thank Dr. Ami Citri (Institute of Life Sciences, The Hebrew University) for his help in setting up the Fluidigm experiment and for his advice. We also thank Dr. Eran Meshorer (Institute of Life Sciences, The Hebrew University) and Dr. Iftach Nachman (Faculty of Life Sciences, Tel Aviv University) for critically reading the manuscript.

### Author contributions

LF and NDR conceived the project and designed the experiments. LF performed the experiments, and NDR performed the data analyses. YK and IM served as advisors on the project, and TP and RS supervised it.

### Conflict of interest

The authors declare that they have no conflict of interest.

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
