## [Review Process File · Molecular Systems Biology]

Regulation of Alternative Splicing at the Single-Cell Level

Lior Faigenbloom, Nimrod D. Rubinstein, Yoel Kloog, Itay Mayrose, Tal Pupko and Reuven Stein

Corresponding author: Reuven Stein, Tel Aviv University

Review timeline:

Submission date:	04 May 2015
Editorial Decision:	17 June 2015
Revision received:	11 October 2015
Accepted:	08 November 2015

Editor: Thomas Lemberger

Transaction Report:

1st Editorial Decision

17 June 2015

Thank you for submitting your work to Molecular Systems Biology. We have now heard back from the three referees who agreed to evaluate your manuscript. As you will see from the reports below, the referees find the topic of your study of potential interest. They raise however several points, which should be convincingly addressed in a revision of this work.

The comments made by the reviewers are very clear in this regard and it will be particularly important to address point 1 of reviewer #1 to ascertain the quality of the data and the correct interpretation of the source of variability observed in your data.

If you feel you can satisfactorily deal with these points and those listed by the referees, you may wish to submit a revised version of your manuscript. Please attach a covering letter giving details of the way in which you have handled each of the points raised by the referees. A revised manuscript will be once again subject to review and you probably understand that we can give you no guarantee at this stage that the eventual outcome will be favorable.

Reviewer #1:

In the manuscript by Faigenbloom, using single cell isoform-specific qRT-PCR, the authors quantified the splicing pattern of 44 alternative exons in single cells from three distinct human cell lines, and correlated the variance of inclusion levels of alternative exons among individual cells with different features. Their results demonstrated that the precision of exon inclusion is mostly determined by the degree of evolutionary conservation of flanking intronic sequences. In addition, the precision is also affected by the inclusion level itself as well as the expression level of the specific transcripts. Finally, based on published single cell RNA sequencing data from human ES

cells, the authors showed that the inclusion of known differentiation-switched cassette exons is also subjected to the same regulation.

This work represents one of the first systematic studies delving deeply into analysis of single cell alternative splicing. Their results advanced our conceptual understanding/appreciation of regulation of alternative splicing at single cell level and may facilitate future mechanistic studies. I believe it is of interest to broad community.

Major points:

My major concern about the manuscript is the data quality.

- 1) The authors did not demonstrate the accuracy of their single cell isoform-specific qRT-PCR. Therefore, the variance they observed among individual cells could be due to biological as well as technical factors. They should for example in silico mix their single cell data and then compare with the data from the bulk samples. In addition, they should repeat their qRT-PCR experiments on total RNAs diluted at the concentration comparable to single cell level and thereby estimate the lower bound of the technical variance. They need to show whether such noise is dependent on the inclusion level and transcript expression level, and how this could affect their conclusion. Similarly, the technical noise of single cell RNA-seq for estimating alternative splicing is totally not clear.
- 2) The number of cells that were used to estimate the variance is not clear. First why did the authors choose 27 top-ranked pre-amplified cDNA samples? How did they determine this number? Second, the authors have applied further several filtering steps to retain samples with measurement of sufficient quality. It is not clear in the end how many cells were used to calculate the variance for each alternative splicing events. If the number is too low, the estimation will be problematic. This is probably the case for single cell RNA-seq data at P0, I am not sure that the bootstrapping performed there is a solution. The authors need to demonstrate the number of cells achieved in this study is enough to draw their conclusion.

Minor points

- 1) In Material and Method, the authors described the motif analysis, but without any results presented.
- 2) Page 7, it is not clear how they performed GO enrichment analysis.
- 3) Page 6, the estimated coefficient should be -0.54 instead of 0.54, or?

Reviewer #2:

Faigenbloom et al used Fluidigm single cell multiplex RT-qPCR technique to analyze the regulation of one type of alternative splicing- inclusion of alternative exons in 293T, MCF7, and U937 cell lines. They analyzed 44 pairs of included and skipped isoforms of RNAs in 81 single cells (27 single cells for each of above three cell types) and found that the accuracy of inclusion percentage of alternative exons is positively correlated with the evolutionary conservation level of the flanking introns of those alternative exons. The accuracy is also positively correlated with the inclusion level (for the minor isoform up to 50%) of the alternative exons and the expression level of the host gene. However, the cell types did not affect the variance of the inclusion level of the alternative exons. Then they used published single cell RNA-seq data to show that the human embryonic stem cells at different passages also show the same patterns of inclusion of alternative exons. The work is interesting but some of the data are not solid enough to support some of their minor conclusions. The paper needs minor revisions and the following issues need to be clearly resolved.

Major points:

1. The authors claim that the cell types did not affect the variance of the inclusion level of the differentiation-switched cassette exons in embryonic stem cells. This is not convincing enough. Since the passage 0 and passage 10 embryonic stem cells are very similar (both are embryonic stem cells although with significant number of genes showing differential expression), so they are not so different types of cells. The author should analyze the variance of the inclusion level of the alternative exons in relatively different cell types, for examples between 8-cell stage embryos and passage 10 embryonic stem cells (or between 4-cell stage embryos and 8-cell stage embryos, etc.). If the cell types still do not affect the variance of the inclusion level of the alternative exons, then their conclusion is convincing.

Minor points:

1. Page #15, Line #6: 'The concentration of each primer pair was chosen as 50uM'. Is this too high as final working concentration? Is it 50nM?
2. Supplemental Fig. 4 & 5: Why do these two figures have background color?

1st Revision - authors' response

11 October 2015

Reviewer #1:

Major points:

My major concern about the manuscript is the data quality.

1) The authors did not demonstrate the accuracy of their single cell isoform-specific qRT-PCR. Therefore, the variance they observed among individual cells could be due to biological as well as technical factors. They should for example in silico mix their single cell data and then compare with the data from the bulk samples.

To address the first point, we first tested how the inclusion levels of the single-cell RT-qPCR samples correspond to those obtained from the bulk RNA samples. To this end, we compared the mean inclusion levels across the single-cell samples with those obtained from the bulk RNA samples (in all the five dilutions performed). No significant difference was observed between the single-cell data and the bulk data, indicating that the inclusion levels in our single-cell experiment are reflective of the inclusion levels in the bulk data.

In addition, to rule out any technical bias between the two FIR conservation groups, which may have been contributed by different primer efficiencies, we analyzed the quantified primer efficiencies of each of the 88 isoforms in each of the three cell types (mentioned in the text on page 17, under "Primer design and calibration"), and found no significant differences in primer efficiencies between the two FIR conservation groups.

We added a paragraph to the Results section (page 6, lines 6-11) describing the results of these two analyses, as well as two figures (Appendix Figs S2C and S2D, respectively), and a description of the analyses in the Materials and Methods section (pages 27 and 17, respectively).

In addition, they should repeat their qRT-PCR experiments on total RNAs diluted at the concentration comparable to single cell level and thereby estimate the lower bound of the technical variance. They need to show whether such noise is dependent on the inclusion level and transcript expression level, and how this could affect their conclusion.

We performed a control experiment in which we measured the expression levels of the 88 isoforms in our data from the same three cell types, on the same microfluidic multiplex RT-qPCR platform. In this experiment, we obtained bulk RNA from total cell population, which was diluted to lower than the average single-cell RNA concentration, as suggested by the reviewer. This should provide a conservative estimation of the technical variance (and hence higher than the expected technical variance in the single-cell RT-qPCR experiment), since we anticipate RNA concentrations to be negatively correlated with technical variance. The data of this control experiment and the single-cell experiment were identically processed.

Our results from this control experiment show that, as expected, the variance of inclusion levels is substantially and significantly higher in the single-cell experiment than in the control experiment. We thus conclude that the variance of inclusion levels in the single-cell experiment reflects genuine cell-to-cell variability.

We added a paragraph to the Results (from page 7 line 10 to page 8 line 9) describing the analyses of this experiment, which are shown in Fig EV1, and a description of the analyses corresponding to these analyses (page 22).

Specifically, we carried out the following analyses to establish this conclusion:

1. We first tested whether the expression levels in the control experiment are indeed lower than in the single-cell experiment, as expected from the above-mentioned dilution. The average expression level in the control experiment was 3.32 fold lower than in the single-cell experiment ($P < 2 \times 10^{-16}$).

This confirms that the expression levels of the different exons in the control experiment are below those of the single-cell experiment. Expression levels are shown in Fig EV1A.

2. We next examined the variance of the inclusion levels in the control experiment. This analysis showed that the average fold change in the variance of inclusion levels between the single-cell and control experiments is 6.36 ($P = 1.03 \times 10^{-7}$). This confirms that the variance of inclusion levels in the single-cell experiment is significantly higher than the technical noise inherent to this experimental setup, suggesting that inclusion-level differences among single cells are genuine and not due to technical noise alone. Inclusion levels are shown in Fig EV1B.

3. We next analyzed the effects of inclusion levels, expression levels, and FIR conservation on the variance of inclusion levels. Assuming a Bernoulli model of exon inclusion in the control-

experiment data, we expect the variance of the inclusion levels to be proportional to $\frac{p(1-p)}{n}$

(where p denotes inclusion level and n denotes expression level of the included and skipped isoforms). Indeed, we observe an inverse relationship between the variance of inclusion levels and dominant inclusion or exclusion levels in the control-experiment data (shown in Fig EV1C). Expression levels were also found to have a significant effect on the variance of inclusion levels ($P = 0.007$ and shown in Fig EV1D). FIR conservation, as expected, was not found to have a significant effect on the variance of inclusion levels ($P = 0.076$).

Similarly, the technical noise of single cell RNA-seq for estimating alternative splicing is totally not clear.

We first subjected the RNA-seq data to quality control using the FastQC software. Then, in our analysis, we eliminated all non-expressed genes. Both of these steps are described in the text (page 22). As these data were not produced as part of this study and did not include bulk RNA data in their original publication, we do not have any direct means of estimating the technical noise inherent to these data. Nevertheless, the agreement between the results of the analysis of our single-cell data and of these hESC data suggest that our conclusions regarding what affects the precision of inclusion levels are valid.

2) The number of cells that were used to estimate the variance is not clear. First why did the authors choose 27 top-ranked pre-amplified cDNA samples? How did they determine this number?

We thank the reviewer for bringing up this point and apologize for our lack of clarity. The Fluidigm microarray contains 96×96 reaction wells. That is, 96 primer pairs amplify 96 samples of cDNA templates. In order to evenly cover 44 pairs of isoforms (i.e., included and skipped isoforms with respect to each cassette exon) from different cell types, we amplified 27 cDNA template samples from each of the three cell types: 293T, MCF7, and U937 (which originate from different tissues), totaling in 81 cDNA samples amplified. We additionally included 3 samples where no cDNA template was used, i.e., negative controls. For the remaining 12 samples, we tried to use cDNA obtained from 100 cells, diluted to a concentration of a single cell, as additional controls but these cells did not lyse properly and were thus discarded. As for primer pairs, we chose included and skipped isoforms corresponding to 22 exons from each FIR conservation group (88 isoforms in total) and eight controls: three primer pairs amplifying housekeeping genes as positive controls loaded in duplicates and no-primer-control as negative controls, also loaded in duplicates (and not in triplicates, as mistakenly stated in the old version of the manuscript and has now been corrected in page 19).

For selecting the 27 single cells (per each cell type), we first analyzed 59 single cells since we were concerned that not all single-cell samples will be appropriate for our experiment (e.g., cell sorting issues which would result with either more than one cell or no cells at all, per well). To verify that the samples that were chosen to the single-cell RT-qPCR assay originated from a single cell, we selected from these 59 single cells only those that showed a similar expression level of three housekeeping genes.

Clarified details are given in the revised version under the “Verification of the amount and quality of single-cell cDNA” header in the Materials and Methods section (page 16).

Second, the authors have applied further several filtering steps to retain samples with measurement of sufficient quality. It is not clear in the end how many cells were used to calculate the variance for each alternative splicing event. If the number is too low, the estimation will be problematic. This is

probably the case for single cell RNA-seq data at P0, I am not sure that the bootstrapping performed there is a solution. The authors need to demonstrate the number of cells achieved in this study is enough to draw their conclusion.

In our filtering steps, any exon which remained with less than three single-cell samples was eliminated so we can reliably estimate the variance of inclusion levels. The final numbers of samples analyzed in both the single-cell RT-qPCR data and the hESC data are provided in Tables EV5 and EV9, respectively. We have also added histograms of the numbers of samples in the RT-qPCR and hESC data sets (Appendix Figures S9A and S9B, respectively).

In the case of the RT-qPCR data, most exons have much higher numbers of samples than three. Notably, in these data the conserved FIR group has a higher number of samples than the non-conserved group. We performed our bootstrap procedure in order to rule out the possibility that the higher variance in inclusion levels of the non-conserved FIR group is merely due to under-estimation resulting from the fact that this group had fewer samples than the conserved FIR group. In our bootstrap procedure we simulated samples of two FIR conservation groups – with numbers and expression levels identical to those in the real data. The inclusion levels were sampled uniformly from a [0,1] interval, for both FIR conservation groups, with an identical over-dispersion factor. Therefore, if differences in the numbers of samples between the two FIR conservation groups are responsible for the significant effect that FIR conservation group has on the variance of inclusion levels, we would expect to observe such an effect in our bootstrap simulated data procedure. Since this was found not to be the case (see pages 25-26), we conclude that the significant effect of FIR conservation group on the variance of inclusion levels is not an artifact of the differences in the number of samples. For the hESC data, such a bootstrap procedure was not performed since cassette exons with low FIR conservation actually have a larger number of samples than those with high FIR conservation. Notwithstanding, even if we analyze hESC cassette exons with at least ten samples (which means using only the P10 population), our results remain essentially unaffected. This is now added to the text on page 26.

Minor points

1) In Material and Method, the authors described the motif analysis, but without any results presented.

We thank the reviewer for requesting clarification on this point. The motif analysis is first discussed in the Introduction section where we mention that conserved FIRs are enriched with binding motifs of known splicing factors and refer to Tables EV2 and EV3, for upstream and downstream FIRs, respectively. In the Materials and Methods section we describe how this analysis was performed and also refer to Tables EV2 and EV3. Since quite a few motifs are enriched, we did not list them in the main document. We hope this clarifies our choice and appropriately addresses the reviewer's comment.

2) Page 7, it is not clear how they performed GO enrichment analysis.

In the revised manuscript we added a paragraph in the Materials and Methods section, under the header: "Gene ontology enrichment analysis of cassette exons with conserved flanking intronic regions" (page 13) describing exactly how this analysis was performed.

3) Page 6, the estimated coefficient should be -0.54 instead of 0.54, or?

We thank the reviewer for catching this mistake. It is now corrected to -0.54.

Reviewer #2:

Major points:

1. The authors claim that the cell types did not affect the variance of the inclusion level of the differentiation-switched cassette exons in embryonic stem cells. This is not convincing enough. Since the passage 0 and passage 10 embryonic stem cells are very similar (both are embryonic stem cells although with significant number of genes showing differential expression), so they are not so different types of cells. The author should analyze the variance of the inclusion level of the alternative exons in relatively different cell types, for examples between 8-cell stage embryos and

passage 10 embryonic stem cells (or between 4-cell stage embryos and 8-cell stage embryos, etc.). If the cell types still do not affect the variance of the inclusion level of the alternative exons, then their conclusion is convincing.

We apologize for the lack of clarity on this point. We did not intend to claim that the P0 and P10 are different cell types (as in the RT-qPCR cell-line data), but rather that these cells represent two populations of the same embryonic state. Since we used these two stem-cell population data sets to determine what affects precision of cassette-exon inclusion levels in stem cells, we wanted to verify that the P10 cells and P0 cells have similar patterns of inclusion levels to exclude the possibility that the P10 cells have begun to differentiate. Our analysis indeed shows that the two cell populations are very similar, at least with respect to the variances of their inclusion levels, without implying that the observed variance of inclusion levels is regulated by different embryonic stages. This is now better described in the revised manuscript (page 10, lines 2-7).

Nevertheless, it is interesting to study the effect of different embryonic stages on the variance of inclusion levels. We therefore assessed the effect of different embryonic stages on the variance of inclusion level, using the stem cell data (P0, P10) with either 8-cells stage RNA-seq data, or both 4-cells stage and 8-cells stage RNA-seq data (both from the same source from which we obtained the stem-cell data - Yan et al. 2013). These analyses did not reveal a significant effect of the embryonic stages in either case. In this respect it should be noted that in our hESC analysis we only chose cassette exons that their inclusion levels were previously shown to largely differ between stem cells and differentiated cells (i.e., differentiation-switch cassette exons, Venables et al, 2013; Gabut et al, 2011; Han et al, 2013; Irimia et al, 2014). The rationale in studying only the differentiation-switch cassette exons is that the regulation of their inclusion levels should be more functionally important than that of cassette exons which their inclusion levels do not vary between stem cells and differentiated cells. However, the importance of this group of exons in earlier embryonic stages has not been studied. In addition, it has been reported that individual cells within 4 and 8 cell embryonic stages are already not homogeneous, as they differ both in the variety and the level of some of the genes they express (e.g., Biase et al. 2014, Piotrowska-Nitsche et al. 2005, Plachta et al. 2011). Given our concerns for the relevance of these early embryonic stages to our analysis, we therefore did not include these analyses in our manuscript.

Minor points:

1. Page #15, Line #6: 'The concentration of each primer pair was chosen as 50uM'. Is this too high as final working concentration? Is it 50nM?

We thank the reviewer for pointing us to this lack of clarity. Each primer pair (50 μ M) was diluted to 500 nM (by combining 2 μ l from each primer pair with 18 μ l of DNA Suspension Buffer) to prepare the $\times 10$ pooled primer Mix. This is now better described in the revised manuscript (page 17).

2. Supplemental Fig. 4 & 5: Why do these two figures have background color?

We have changed the background of these figures (now Appendix Figs S4 and s5) to white.

References

- Biase FH, Cao X, Zhong S. 2014. Cell fate inclination within 2-cell and 4-cell mouse embryos revealed by single-cell RNA sequencing. *Genome Res.* 24(11):1787–96
- Piotrowska-Nitsche K, Perea-Gomez A, Haraguchi S, Zernicka-Goetz M. 2005. Four-cell stage mouse blastomeres have different developmental properties. *Development.* 132:479–90
- Plachta N, Bollenbach T, Pease S, Fraser SE, Pantazis P. 2011. Oct4 kinetics predict cell lineage patterning in the early mammalian embryo. *Nat. Cell Biol.* 13(2):117–23
- Yan L, Yang M, Guo H, Yang L, Wu J, et al. 2013. Single-cell RNA-Seq profiling of human preimplantation embryos and embryonic stem cells. *Nat. Struct. Mol. Biol.* 20(9):1131–39

2nd Editorial Decision

08 November 2015

Thank you again for sending us your revised manuscript. We are now satisfied with the modifications made and I am pleased to inform you that your paper has been accepted for publication.

Reviewer #1:

The revision has adequately addressed my concerns.

Reviewer #2:

The authors have addressed all the questions I raised. If the journal has enough publication space, it should be accepted now.